# Bipolar disorder among married women in Bangladesh: Survey in Rajshahi city

**Md. Abdul Wadood**[1]☉, **Md. Rezaul Karim**[2‡], **Abdullah Al Mamun Hussain**[3‡], **Md. Masud Rana**[4‡], **Md. Golam Hossain**[5]☉*

1 Medical Centre, University of Rajshahi, Rajshahi, Bangladesh, 2 Department of Biochemistry and Molecular Biology, University of Rajshahi, Rajshahi, Bangladesh, 3 Department of Psychiatry, Rajshahi Medical College, Rajshahi, Bangladesh, 4 Department of Population Science and Human Resource Development, University of Rajshahi, Rajshahi, Bangladesh, 5 Health Research Group, Department of Statistics, University of Rajshahi, Rajshahi, Bangladesh

☉ These authors contributed equally to this work.
‡ These authors also contributed equally to this work.
* hossain95@yahoo.com

**Data Availability Statement:** All relevant data is found in the manuscript and supporting information files

**Funding:** The authors received no specific funding for this work.

## Abstract

### Background

Bipolar disorder (BPD) is a major mental disorder which not only affects the personal and social functioning of an individual, but also inflicts a huge economic burden on the family. Yet, the study of BPD in Bangladesh is rare and poorly documented. Responding to the dire need, we conducted a new study to determine the prevalence of, and detect the associated factors of, BPD among married women in Rajshahi City, Bangladesh.

### Methods

We conducted a cross-sectional study, selecting households in Rajshahi City using a multi-stage random sampling design. The data consisted of 279 married women, who were screened for BPD using the bipolar spectrum diagnostic scale (BSDS). Frequency distribution, chi-square test and binary logistic regression model were used respectively to determine the prevalence, identify the associated factors and quantify their effects on BPD.

### Results

The prevalence of BPD among married women in Rajshahi City was 2.5%, with an additional 7.2% classified as probable BPD. A binary logistic regression analysis established the following six main factors of BPD: (1) comorbid mental disorder [AOR = 8.232, 95% CI = (1.397, 50.000), p<0.05]; (2) poor relationship with husband [AOR = 11.775, 95% CI = (2.070, 66.667), p<0.01]; (3) poverty [AOR = 1.600, 95% CI = (2.086, 122.709), p<0.01]; (4) high educational level [AOR = 0.177, 95% CI = (0.037, 0.843), p<0.05]; (5) lack of immediate treatment if sick [AOR = 2.941, 95% CI = (1.259, 6.871), p<0.05]; and (6) death of beloved one/s [AOR = 2.768, 95% CI = (1.130, 6.777), p<0.05].

**Competing interests:** The authors have declared that no competing interests exist

## Limitations

Our survey involved self-reporting, which is typically affected by differing levels of understanding and bias. Also, a cross-sectional, observational study cannot establish an actual cause-and-effect relationship. Some other potentially important factors such as environment, lifestyle, familial customs, effects of drugs, treatment options and outcomes, etc. were not studied. Also, save mental disorder, all other comorbidities remained undocumented.

## Conclusion

This study laid down the foundation for conducting further research on identifying different factors affecting BPD, and for studying other issues related to BPD among married women in Bangladesh. Among such factors are familial environment and culture, comorbidities, treatment options, treatment outcomes, biochemical feature, environmental factors, etc. This study also recommends that, while treating BPD patients, health professionals should focus on comorbidities and family matters.

## Introduction

Bipolar disorder (BPD) is a major psychiatric illness characterized by fluctuations of mood. It disrupts the patient's personal and social life; and it inflicts a huge economic burden on the family. Its prevalence varies between 0.2% and 6% [1–5] across different countries. In the bipolar spectrum form, its prevalence ranges from 2.6% to 7.8% [6]. A recent study conducted in 11 countries—mainly in the Americas, Europe and Asia—found a lifetime BPD prevalence of 2.4% [7]. BPD is the ninth leading cause of years-of-healthy-life-lost due to premature mortality and disability [8]. In 2004, BPD affected an estimated 29.5 million people worldwide [8]; and an estimated 0.9% of the total global burden of disease was attributed to BPD. In 2013, BPD accounted for 9.9 million disability-adjusted life-years (DALYs), or 0.4% of total DALYs and 1.3% of total years lived with disability [9].

BPD is a heritable illness [10, 11]. The first-degree relatives of a patient show a significantly higher rate of mood disorder and social cognitive deficits [12]. BPD prevalence also has a significant positive relationship with hypertension, dyslipidemia and diabetes [13]. Environmental factors influence its severity and clinical course [14]. Stressful life events, both in childhood and in adulthood, and alcohol or substance abuse affect the onset, recurrence and severity of BPD [15, 16]. Its incidence is increased by viral infection, substance abuse and trauma [17]. Common mental comorbidities of BPD include anxiety, substance abuse, conduct disorders, eating disorders, abnormal sexual behavior, attention-deficit/hyperactivity, impulse control, autism spectrum disorders, etc. Medical comorbidities are migraine, thyroid illness, obesity, type II diabetes and cardiovascular diseases [18].

BPD starts at age 18–22 years. It is common in both males and females, though the course of the disease in the two gender groups differs [19, 20]. Usually, women show a predominance of depression and mixed mania; and they commonly develop it at an older age with one or more physical comorbidities [21].

In Bangladesh, there have been some studies on general mental disorders. A few of these studies also investigated, though in a small scale, the prevalence of clinically diagnosed BPD along with other mental disorders [22, 23, 24]. Some other studies have been conducted on specific mental disorders such as depression [25, 26], schizophrenia [27], anxiety disorder

[28], substance abuse disorder [29, 30], and obsessive-compulsive disorder [31]. To the best of our knowledge, no study has been done in Bangladesh exclusively on BPD—neither in the general population, nor among married women.

BPD causes long-lasting adverse effects on psycho-social functioning of the individual, and it generates negative financial implications causing intense suffering for the diseased individual. Moreover, in Bangladesh, wives and mothers play a major role in doing household works and rearing children. If they become sick, they cannot perform the duties efficiently. Consequently, the family is hurled into an abyss of suffering, and the country faces a great public health concern.

In light of above-mentioned adversities, our study was aimed at determining the prevalence of BPD among married women in Rajshahi City, Bangladesh, and at identifying the associated risk factors and quantifying their effects on BPD.

## Materials and methods

We conducted a cross-sectional household study. All households in Rajshahi City constituted the population for this survey. We shall explain the sampling design in the next two paragraphs. From each selected household, one married woman was invited to respond to the survey. All selected married women were currently living with their husbands. The respondents' age ranged from 15 to 82 years.

### Sample size determination

Rajshahi is one of the four big cities of Bangladesh having an area of 97.18 sq. km. and a total population of 4,48,087 [32]. The city is divided into 30 wards, which are further subdivided into muhallas (neighborhoods), and consist of 99,222 households. The mathematical formula $n = N/(1+Nd^2)$ was used to determine the sample size for this study [33], where n = sample size, N = population size and d = margin of error. Choosing d = 0.05, the formula indicated that n = 398 would suffice for this study. Assuming a 90% rate of response from the selected women, initially 450 married women were selected to participate in the study.

### Sampling method

The survey participants were selected using a multistage random sampling (Fig 1). In the first stage, three wards were randomly selected out of the 30 wards of Rajshahi City, using a probability proportional to size sampling scheme. In the second stage, three muhallas were selected from each chosen ward by random sampling, again using a probability proportional to size sampling scheme. In the third stage, 50 households were selected from each chosen muhalla, using a simple random sampling. If the chosen household had only one married woman, she was invited to participate in the survey; otherwise, only one of the married women was chosen at random. All information about the number of households within each muhalla was collected from Rajshahi City Corporation Office. The randomizations were implemented by senior researchers.

### Ethical approval

Before collecting data, ethical clearance for the study was taken from the Institutional Animal, Medical Ethics, Biosafety and Biosecurity Committee (IAMEBBC) for Experimentation on Animal, Human, Microbes and Living Natural Sources, Institute of Biological Sciences, University of Rajshahi, Bangladesh (Memo No: 120/ 320/ IAMEBBC/ IBSc, dated 11 April, 2019).

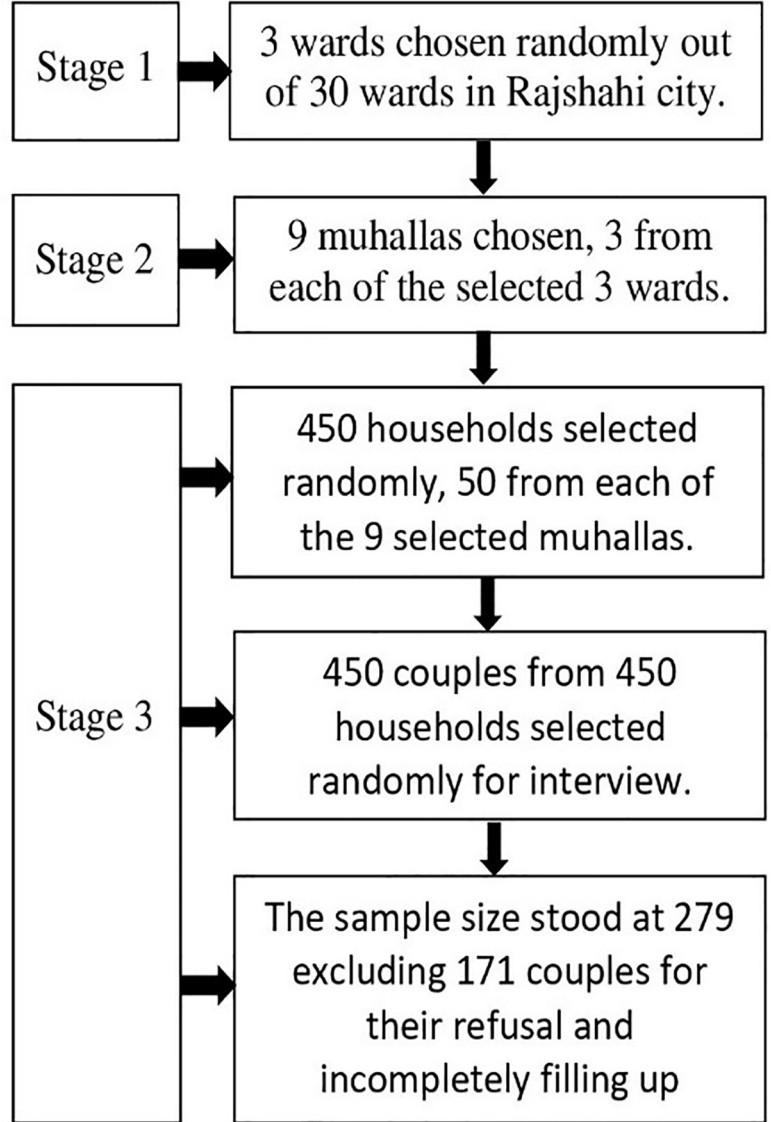

**Fig 1. A three-stage sampling method was used to include households/married women in the survey.**

## Data collection

For data collection, we used a semi-structured questionnaire, which was originally written in English and later translated into Bangla to help participants understand it easily. The first author prepared the first draft of the translation; subsequently, other authors reviewed and improved it. The final version of the questionnaire included the 20 questions (Questions 15–34) of the bipolar spectrum disorder scale (BSDS).

Three teams were trained to collect data; and one team was assigned to each of the three selected wards of the city. Each team consisted of one male and one female postgraduate student of the Department of Statistics, University of Rajshahi. The interviewers discussed the details of the research with the participants. A total of 96 women (slightly over 20%) declined to give any information. The remaining 354 agreed to provide information, and their written consent was taken. For respondents under 18 years old, consent of their guardians (family

heads) was also taken (incidentally, although the legal minimum age of marriage in Bangladesh is 18 years for females, occurrence of child marriage is still very high [34, 35]). Fifty three responding women took the questionnaire with them, and asked the interviewers to collect it another day. Among them, 16 women failed to return their questionnaires. The survey was conducted at the respondents' place of choice during the period May 15 through July 30, 2019.

When data were entered in a spread sheet, we detected some (59 married women) responses had one or more missing values; consequently, we excluded these respondents. Finally, complete responses from n = 279 married women were available for analysis. Thus, the achieved sample size fell short of the desired size of 398. Accordingly, the margin of error increased to d = 0.06.

## Outcome variable

The outcome variable, BPD in married women, was determined using BSDS, which is an effective tool with sensitivity 0.76 and specificity 0.93 [36]. A study compared the diagnostic accuracy of several screening tools, and found that BSDS had the highest reliability (0.83) [37]. The BSDS was also used in developing countries such as Iran [38]. The score of points of the 20 questions ranged from 0 to 25 [39]. In this study, we classified our sample into three classes such as (i) no bipolar disorder (0–12 points), (ii) probable BPD (13–19 points) and (iii) BPD (20–25 points). However, as the prevalence of probable BPD and BPD were very low, these two classes were merged into one, and the combined class was simply called 'BPD'. Thus, our respondents were classified into two categories, which were used for chi-square tests and binary logistic regression model.

## Independent variables

Based on similar studies conducted in the past and keeping in view the objectives of our study, some socio-economic, demographic, anthropometric, familial and psycho-social factors were considered as independent variables in this study. The 23 independent variables were: age group, nutritional status, religion, respondent's and their parents educational level, respondent's occupation, type of family, number of family members, family's monthly income, age at first marriage, duration of present conjugal life, miscarriage/abortions, death of children, number of children alive, number of marriage, comorbid stress/anxiety, relationship with husband, if sick treated immediately, comorbid chronic disease, family members' chronic disease, comorbid mental disorder, blood relative's mental disorder, and death of beloved one/s.

## Statistical analysis

A frequency distribution was used to determine BPD prevalence. Chi-square test and binary logistic regression model were used respectively to detect associated significant factors and to measure their effects on BPD among married women in Rajshahi City, Bangladesh. The software SPSS (IBM, version 22) was used to analyze the data.

## Results

The frequency distribution revealed that the prevalence of BPD, probable BPD and no BPD among married women in Rajshahi city, Bangladesh were 2.5%, 7.2% and 90.3% respectively (Fig 2).

Chi-square ($\chi^2$) tests identified the following ten variables as statistically significant factors associated with BPD among married women (Table 1): respondent's education level, family's monthly income, age at the first marriage, relationship with husband, if sick treated

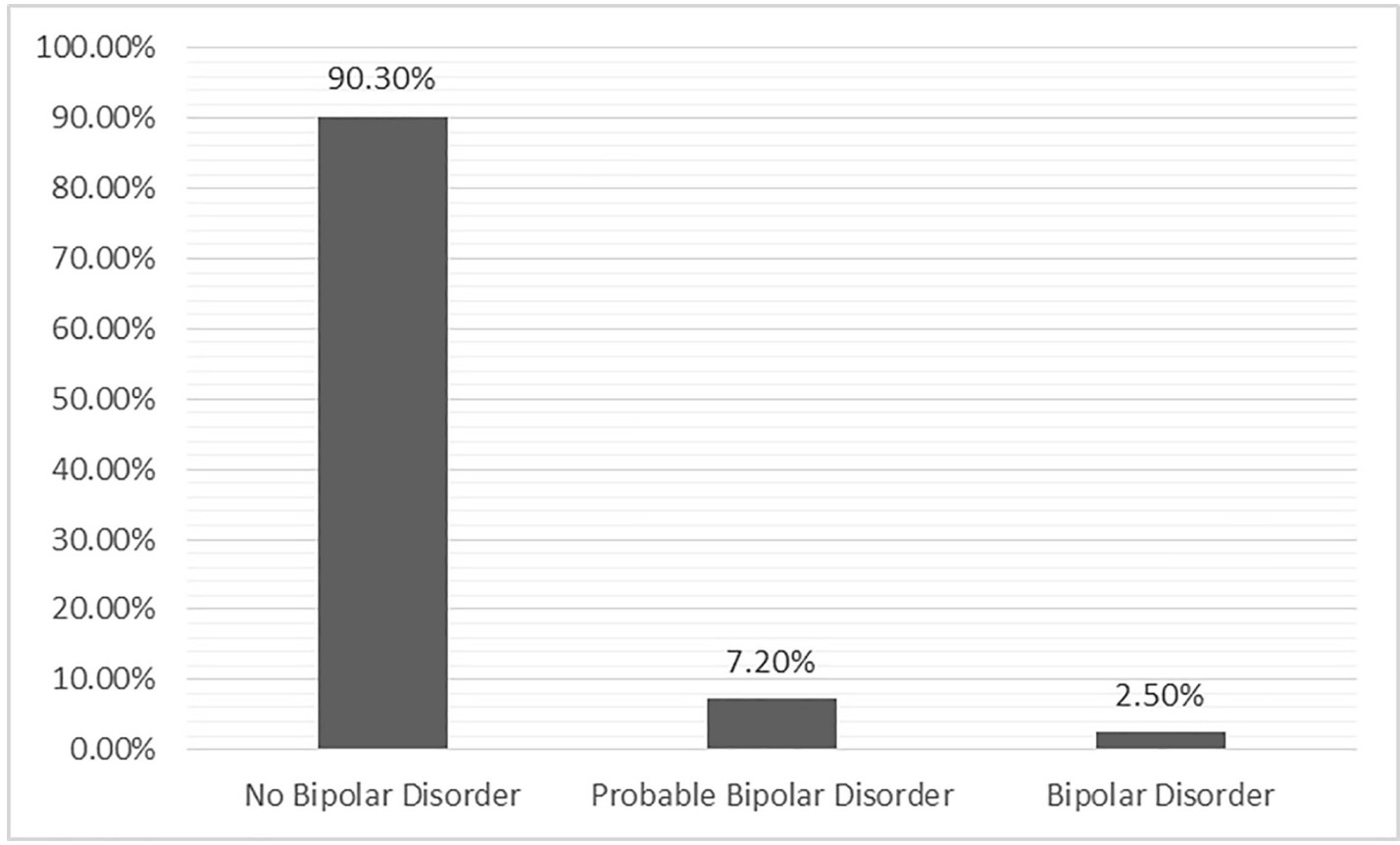

**Fig 2. Prevalence of BPD among married women in Rajshahi city, Bangladesh.**

immediately, comorbid chronic disease, family members' chronic disease, comorbid mental disorder, blood relative's mental disorder and death of beloved one/s (Table 1).

Only significant factors (detected by chi-square test) were included as independent variables in the logistic model. Table 2 shows the results of the binary logistic regression model, which pronounced six of the ten factors as statistically significant in altering the odds of developing BPD. Women with comorbid mental disorder had eight times higher odds to develop BPD [AOR = 8.323, 95% CI = (1.397, 50.000), p<0.05]. Women having poor relationships with their husbands had twelve times higher odds to have the disorder [AOR = 11.775, 95% CI = (2.070, 66.667), p<0.01]. Women coming from poor families were 16 times more vulnerable to develop BPD than those from rich families [AOR = 16.000, 95% CI = (2.086, 122.709), p<0.01]. Surprisingly, women with higher education had six times higher odds to have BPD than women with only primary education [AOR = 0.177, 95% CI = (0.037, 0.843), p<0.05]. Women who were not treated immediately if sick showed about three times more chance to develop BPD [AOR = 2.941, 95% CI = (1.259, 6.871), p<0.05]. Women whose beloved one/s died were about three times more vulnerable to have BPD than those who did not lose their dear ones [AOR = 2.768, 95% CI = (1.130, 6.777), p<0.05] (Table 2).

## Discussion

Our study aimed at determining the prevalence of bipolar disorder and its associated factors among married women in Bangladesh. For this purpose, a survey was conducted in Rajshahi

**Table 1. Chi-square test identify ten socio-economic, familial and psychological factors that have significant association with BPD among married women in Rajshahi City, Bangladesh.**

| Variables N (%) | No BPD N (%) | BPD N (%) | $\chi^2$-value | p-value |
|---|---|---|---|---|
| **Respondent's education level** | | | 7.029 | 0.040 |
| Uneducated, 37 (13.3) | 35 (94.6) | 2 (5.4) | | |
| Primary, 98 (35.1) | 92 (93.9) | 6 (6.1) | | |
| Secondary, 90 (32.3) | 81 (90.0) | 9 (10.0) | | |
| Higher, 54 (19.3) | 44 (81.5) | 10 (18.5) | | |
| **Family's monthly income** | | | 12.679 | 0.002 |
| Poor, 84 (30.1) | 68 (81.0) | 16 (19.0) | | |
| Middle, 133 (47.7) | 124 (93.2) | 9 (6.8) | | |
| Rich, 62 (22.2) | 60 (96.8) | 2 (3.2) | | |
| **Age at the first marriage** | | | 6.732 | 0.013 |
| <18 years, 117 (41.9) | 112 (95.7) | 5 (4.3) | | |
| ≥18 years, 162 (58.1) | 140 (86.4) | 22 (13.6) | | |
| **Relationship with husband** | | | 11.178 | 0.001 |
| Good, 103 (37.0) | 101 (98.1) | 2 (1.9) | | |
| Poor, 176 (63.0) | 151 (85.8) | 25 (14.2) | | |
| **If sick treated immediately** | | | 6.673 | 0.017 |
| No, 52 (18.6) | 42 (80.8) | 10 (19.2) | | |
| Yes, 227 (81.4) | 210 (92.5) | 17 (7.5) | | |
| **Comorbid chronic disease** | | | 14.938 | 0.001 |
| No, 150, (53.7) | 145 (96.7) | 5 (3.3) | | |
| Yes, 129 (46.3) | 107 (82.9) | 22 (17.1) | | |
| **Family members' chronic disease** | | | 15.112 | 0.001 |
| No, 178 (63.8) | 170 (95.5) | 8 (4.5) | | |
| Yes, 101 (36.2) | 82 (81.2) | 19 (18.8) | | |
| **Comorbid mental disorder** | | | 39.912 | 0.001 |
| No, 198 (71.0) | 193 (97.5) | 5 (2.5) | | |
| Yes, 81 (29.0) | 59 (72.8) | 22 (27.2) | | |
| **Blood relative's mental disorder** | | | 10.178 | 0.004 |
| No, 235 (84.2) | 218 (92.8) | 17 (7.2) | | |
| Yes, 44, (15.8) | 34 (77.3) | 10 (22.7) | | |
| **Death of beloved one/s** | | | 5.307 | 0.025 |
| No, 131 (47.0) | 124 (94.7) | 7 (5.3) | | |
| Yes, 148 (53.0) | 128 (86.5) | 20 (13.5) | | |

City, Bangladesh. The prevalence of BPD found in this study was 2.5%, compared to 2.4% in 11 countries of the Americas, Europe, and Asia [7], 2.0% in England [40], 2.2% in Canada [41] and 1.2% in Singapore [42]. In our study, the prevalence of probable BPD was found to be 7.2%. Hence, the prevalence of BPD ranged from 2.5% to 9.7%, which is consistent with a global prevalence of 2.6% to 7.8% [6]. In comparison, the prevalence of BPD was estimated to be 8.6% in India [43] and 14.3% in Pakistan [44]. BPD prevalence is usually higher in urban environments than in rural areas [45]. This might also be a cause of the comparatively higher prevalence of BPD in our study, as all of our subjects came from urban areas. Such dissimilar findings necessitates conducting more studies either using the same scale and strategy, or using different scales and strategies.

Our study revealed that the women with comorbid mental disorder were eight times more prone to develop BPD, which is comparable to that found in other studies in Europe [3],

**Table 2. Effect of socio-economic, familial and psychological factors on BPD among married women in Rajshahi City, Bangladesh.**

| Variable | B | SE | p-value | AOR* | 95% Cl** of AOR | |
|---|---|---|---|---|---|---|
| | | | | | Lower | Upper |
| **Comorbid mental disorder** | | | | | | |
| No vs Yes[R] | -2.119 | 0.910 | 0.020 | 0.120 | 0.020 | 0.716 |
| **Relationship with husband** | | | | | | |
| Good vs Poor[R] | -2.466 | 0.887 | 0.005 | 0.085 | 0.015 | 0.483 |
| **Family's monthly income** | | | | | | |
| Poor vs Rich[R] | 2.773 | 1.039 | 0.008 | 16.000 | 2.086 | 122.709 |
| Middle vs Rich[R] | 1.266 | 0.996 | 0.204 | 3.546 | 0.503 | 24.994 |
| **Respondent's education level** | | | | | | |
| Uneducated vs Higher[R] | -1.524 | 1.073 | 0.155 | 0.218 | 0.027 | 1.783 |
| Primary vs Higher[R] | -1.730 | 0.796 | 0.030 | 0.177 | 0.037 | 0.843 |
| Secondary vs Higher[R] | -0.579 | 0.679 | 0.394 | 0.561 | 0.148 | 2.120 |
| **If sick, treated immediately** | | | | | | |
| No vs Yes[R] | 1.079 | 0.433 | 0.013 | 2.941 | 1.259 | 6.871 |
| **Death of beloved one/s** | | | | | | |
| No vs Yes[R] | -1.018 | 0.457 | 0.026 | 2.768 | 1.130 | 6.777 |

R- Reference factor

*AOR- Adjusted Odds Ratio

**CI- Confidence Interval.

United States [46] and the entire globe [47]. In this study, we also found that poor relationship with husband was an important factor affecting BPD among married women. Marital life sometimes becomes stressful and can trigger onset or relapse of mental illness such as BPD; on the other hand, marriage can also protect couples from mental disorders [48]. Hence, relationship with husband is a crucial issue: poor relation can either create or trigger mental disturbances; good relation can prevent mental illness. Nonetheless, we could not compare our finding with other studies as no other study included husband-wife relationship as a study variable.

Women from poor families were found more vulnerable to develop BPD than those from rich families, probably because poverty exposed the poor women continually to insecurity, anxiety and stress. A US study agreed that people with low family income were more vulnerable to BPD [49]. Another study observed that adult women of low socioeconomic status had twice the chance of developing mood disorders compared to middle- and high income groups [50].

Surprisingly, our study revealed that women with high education were more likely to have BPD than those with only primary education. A probable explanation for this may be that women with high education suffered from despair for not getting due recognition, power and honor within the family or in society—privilege their high educational status should have earned them. In fact, our observation matches a finding that BPD patients showed a higher likelihood to complete the highest level of education compared to their normal relatives [51]. On the other hand, a Norwegian study found that the association between educational level and BPD prevalence was not statistically significant, although social and occupational functioning was lower among BPD patients compared to healthy ones [52]. Furthermore, in our study, occurrence of BPD among uneducated women was not significantly different from that among women with higher education. Hence, we could not make any conclusive statement regarding the relationship between educational level and BPD occurrence.

Our study found that women who were deprived of getting immediate treatment if sick had a three times higher odds of developing BPD. No other study is available to compare this finding. We can say that such a situation probably breeds a sense of insecurity, agitation and irritation in these women.

Death of beloved one/s was found to be an important risk factor of BPD among married women. This issue is poorly documented in the literature: A study in Denmark found that parental death, especially maternal, increased the chance of BPD in their offspring [45]. Death of dear ones imprints on the human mind a long-lasting psychological effect; and that may be an explanation behind our finding.

Our study determined, for the first time in Bangladesh, the prevalence of bipolar disorder among married women; and it successfully identified some associated significant factors. However, this study also had some limitations. The self-reported responses to the BSDS questionnaire, being dependent predominantly on the respondents' perceptions, may have been affected by differing levels of understanding and bias. Moreover, the cross-sectional observational study could not detect any actual cause-and-effect relationship. Comorbidities (chronic and mental disorders of the subjects and their blood-relatives) could not be accounted for. Also, some other important issues such as environmental factors, lifestyle, familial customs, effects of drugs, treatment options and outcomes, etc. could not be studied. Recognition of these limitations ought to propel the scientific community to implement new, more in-depth and elaborate research strategies.

## Conclusions

The current study determined the prevalence of bipolar disorder and detected some associated risk factors of BPD among married women in Rajshahi City, Bangladesh. We found that 2.5% and 7.2% married women were suffering from BPD and probable BPD respectively. Among the significant risk factors were mental disorder, poor relation with husband, poverty, high educational level, lack of immediate treatment if sick and death of beloved one/s. As no other study on BPD has been conducted in Bangladesh, this current study has laid the foundation for further research regarding different aspects of BPD such as familial environment and culture, comorbidities, treatment options, treatment outcomes, biochemical picture, environmental factors, etc. Furthermore, based on our study, we recommend that health professionals focus on comorbidities and family matters while providing treatment and rehabilitation services to BPD patients. Government authorities and concerned nongovernmental and social organizations should take adequate steps to reduce repression on women and work for ensuring their rights and empowerment both at the household level and in society.

## Supporting information

**S1 Data.**
(SAV)

## Acknowledgments

The authors would like to express their heartfelt thanks to all who participated in or otherwise contributed to this research. The authors also like to thank Professor Jyoti Sarkar, Department of Mathematical Sciences, School of Science, Indiana University Purdue University Indianapolis for improvement of language in this manuscript.

## Author Contributions

**Conceptualization:** Md. Abdul Wadood, Md. Golam Hossain.

**Data curation:** Md. Abdul Wadood, Md. Rezaul Karim, Md. Golam Hossain.

**Formal analysis:** Md. Abdul Wadood, Abdullah Al Mamun Hussain, Md. Masud Rana, Md. Golam Hossain.

**Methodology:** Md. Abdul Wadood, Md. Masud Rana, Md. Golam Hossain.

**Writing – original draft:** Md. Abdul Wadood.

**Writing – review & editing:** Md. Abdul Wadood, Md. Rezaul Karim, Abdullah Al Mamun Hussain, Md. Golam Hossain.

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
