## [Decision Letter · Decision Letter 0]

3 Dec 2019

PONE-D-19-24745

Bipolar disorder among married women in Bangladesh: Survey in Rajshahi city

PLOS ONE

Dear Prof. Hossain,

Thank you for submitting your manuscript to PLOS ONE. After careful consideration, we feel that it has merit but does not fully meet PLOS ONE’s publication criteria as it currently stands. Therefore, we invite you to submit a revised version of the manuscript that addresses the points raised during the review process.

We would appreciate receiving your revised manuscript by Jan 17 2020 11:59PM. To enhance the reproducibility of your results, we recommend that if applicable you deposit your laboratory protocols in protocols.io, where a protocol can be assigned its own identifier (DOI) such that it can be cited independently in the future. For instructions see: http://journals.plos.org/plosone/s/submission-guidelines#loc-laboratory-protocols

We look forward to receiving your revised manuscript.

Kind regards,

Nagendra P. Luitel, MA

Academic Editor

PLOS ONE

Journal Requirements:

'No fund was received from any individual or organization for this study.'

'The funders had no role in study design, data collection and analysis, decision to

publish, or preparation of the manuscript'

4. Your ethics statement must appear in the Methods section of your manuscript. If your ethics statement is written in any section besides the Methods, please move it to the Methods section and delete it from any other section. Please also ensure that your ethics statement is included in your manuscript, as the ethics section of your online submission will not be published alongside your manuscript.

Additional Editor Comments (if provided):

1) Please explain in detail how the randomization (in each step) was done. Who did the randomization i.e. the field workers or the senior researchers?

2) Please explain translation and adaptation process of the Bipolar Spectrum Diagnostic Scale (BSDS) in detail. Was any guideline or checklists used for translation and adaptation of the BSDS in local language? Who were involved in translation and adaptation process?

3) Please include more references in the methods section (under outcome measure sub-section) on the use of BSDS in community survey

4) Please provide some background information (i.e. if they are different than the study sample) of the 59 cases which were discarded from the analysis

5) Finally, as a significant number of participants are below the age of 18, it would be helpful for the readers if you could also provide information about minimum legal age of marriage in Bangladesh

Reviewers' comments:

Reviewer's Responses to Questions

**Comments to the Author**

1. Is the manuscript technically sound, and do the data support the conclusions?

Reviewer #1: No

Reviewer #2: Partly

2. Has the statistical analysis been performed appropriately and rigorously? 

Reviewer #1: No

Reviewer #2: No

3. Have the authors made all data underlying the findings in their manuscript fully available?

Reviewer #1: Yes

Reviewer #2: Yes

4. Is the manuscript presented in an intelligible fashion and written in standard English?

Reviewer #1: No

Reviewer #2: Yes

5. Review Comments to the Author

Reviewer #1: This article examines the prevalence of bipolar disorder in married Bangladeshi women in a particular geographical area.

The manuscript suffers from several flaws that limit its scientific validity and applicability.

1. First, the authors have used a screening instrument that is, by definition, meant to pick up "spectrum disorders" rather than rigorously defined bipolar disorder. Hence, the "prevalence" estimate that they provide is of limited use. All "screen positive" cases should have been confirmed using standard diagnostic criteria.

2. Secondly, the authors have not confirmed if this instrument is valid in non-Western cultures. A perusal of the scale shows that it can easily be misunderstood and that some of the terms in it are hard to translate.

3. It is tautological (and more than a little foolish) to claim that "mental illness" increases the risk for bipolar disorder. Bipolar disorder is, itself, a mental illness! If the authors mean "comorbid mental illness" (such as an anxiety disorder) then this must be mentioned using appropriate terms. Otherwise, it is a serious error which reflects badly on the credibility of the researchers.

4. Similarly, the authors have used simplistic, non-standard terminology ("rich" vs "poor" for income, "good" vs "poor" for quality of the marital relationship) that makes the data analyses difficult to interpret.

5. The writing style and grammar are poor, and the article would benefit from the assistance of a writer skilled in scientific writing in English.

6. Finally, there are serious ethical concerns about the inclusion of victims of child marriage (age 15) in a study of this sort. By including such subjects and accepting such a situation as normal, the researchers are tacitly endorsing the aberrant situation that exists in their country (https://www.independent.co.uk/news/world/asia/bangladesh-child-marriage-law-minimum-age-zero-reduce-baby-marital-unicef-un-a7619051.html) rather than advocating for the rights of vulnerable populations, as any ethical researcher would.

For these reasons, it is my opinion that this article does not merit publication.

Reviewer #2: Minor corrections:

Abstract:

1. Better to use the term “mental disorder” rather than using “mental disease”

Introduction:

2. In place of the term “victim” use “patient”

Methodology

3. Authors have mentioned randomization was done at all three stages. It would be appreciated if they mention what method of randomization was used. A brief description of the process would help readers as well as future researchers.

4. What does “mental position” mean in line 122

5. Regarding the questionnaire, it is not clear whether the 66 items semi structured questionnaire had 20 items of BSDS or it was separately administered. Please clarify

6. For data collection you have mentioned “literate adults” were used as data collectors. I think they should have some level of education to collect data.

7. For female less than 18 years please mention if assent was taken.

8. Grammatical error “All the information of the answered questionnaires entered in computer and coded”

Major comments:

9. In the methodology result and discussion section the authors have looks into different factors (mainly social) and have concluded the precipitating factors, vulnerability. However the factors may have just association rather than direct precipitation. The authors have completely missed the biological and psychological factors. If not studied these could be the domains for discussion.

10. Table 1: the factors like family members’ chronic illness have shown significant difference in BPD and normal groups. Similarly variable “sick treated immediately” makes no sense. This might be misleading as there is no any rational pathway for this. Hence I would recommend to take literature review in consideration before doing statistical tests and making an inference.

11. How were the reference factor selected ? (table 2)

12. My main concern is how were the study variables selected to make inferences?

6. PLOS authors have the option to publish the peer review history of their article (what does this mean?). If published, this will include your full peer review and any attached files.

Reviewer #1: No

Reviewer #2: Yes: Dr. Pawan Sharma

---

## [Author Response · Author response to Decision Letter 0]

7 Jan 2020

Response to Editor and Reviewers 

Journal Name: PLOS ONE 

Tracking No. (Manuscript ID): PONE-D-19-24745

Manuscript Title: “Bipolar disorder among married women in Bangladesh: Survey in Rajshahi city"

Dear Editor,

Thank you very much for providing your and reviewers’ comments on our manuscript. We have modified and revised the manuscript accordingly, and detailed corrections point-by-point is given below:

Editor Comments:

Author’s response: Thank you very much for your useful comments on our manuscript. We have provided our data. Our data in SPSS file, we have provided data as supplementary file (data.sav). 

'No fund was received from any individual or organization for this study.'

'The funders had no role in study design, data collection and analysis, decision to publish, or preparation of the manuscript'

Author’s response: We have revised accordingly.

4. Your ethics statement must appear in the Methods section of your manuscript. If your ethics statement is written in any section besides the Methods, please move it to the Methods section and delete it from any other section. Please also ensure that your ethics statement is included in your manuscript, as the ethics section of your online submission will not be published alongside your manuscript.

Author’s response: The ethical statement has been written in the materials and methods section [Page 6, Line: 125-130], and deleted from other place. 

Additional Editor Comments (if provided):

1) Please explain in detail how the randomization (in each step) was done. Who did the randomization i.e. the field workers or the senior researchers?

Author’s response: We have mentioned in detail about the randomization of our subject [Page 6, Line 115-124]

2) Please and explain translation adaptation process of the Bipolar Spectrum Diagnostic Scale (BSDS) in detail. Was any guideline or checklists used for translation and adaptation of the BSDS in local language? Who were involved in translation and adaptation process?

Author’s response: The translation adaptation process has been described in the manuscript [Page 6-7, Line 131-135]

3) Please include more references in the methods section (under outcome measure sub-section) on the use of BSDS in community survey

Author’s response: As per your suggestion, some relevant and necessary references have been included regarding use of BSDS [Page 7-8, Line 153-162].

4) Please provide some background information (i.e. if they are different than the study sample) of the 59 cases which were discarded from the analysis

Author’s response: This issue has been explained [Page 7, Line 140-152]. 

5) Finally, as a significant number of participants are below the age of 18, it would be helpful for the readers if you could also provide information about minimum legal age of marriage in Bangladesh

Author’s response: As per your advice, minimum legal age of marriage and real situation was described with reference [Page 7, Line 142-149]. 

Reviewers' comments:

Comments to the Author

1. Is the manuscript technically sound, and do the data support the conclusions?

Reviewer #1: No

Reviewer #2: Partly

2. Has the statistical analysis been performed appropriately and rigorously?

Reviewer #1: No

Reviewer #2: No

 3. Have the authors made all data underlying the findings in their manuscript fully available?

Reviewer #1: Yes

Reviewer #2: Yes

 4. Is the manuscript presented in an intelligible fashion and written in standard English?

Reviewer #1: No

Reviewer #2: Yes

5. Review Comments to the Author

Reviewer #1: This article examines the prevalence of bipolar disorder in married Bangladeshi women in a particular geographical area.

The manuscript suffers from several flaws that limit its scientific validity and applicability.

1. First, the authors have used a screening instrument that is, by definition, meant to pick up "spectrum disorders" rather than rigorously defined bipolar disorder. Hence, the "prevalence" estimate that they provide is of limited use. All "screen positive" cases should have been confirmed using standard diagnostic criteria.

Author’s response: Thank you for your comments. The scientific validity and specificity of BSDS in screening bipolar disorder have been discussed with references [Page 7-8, Line 153-171].

2. Secondly, the authors have not confirmed if this instrument is valid in non-Western cultures. A perusal of the scale shows that it can easily be misunderstood and that some of the terms in it are hard to translate.

Author’s response: The validity of BSDS in non-Western community was added in Page 7-8, Line 153-171. The translation adaptation process has been described in the manuscript [Page 6-7 Line 131-137]. 

3. It is tautological (and more than a little foolish) to claim that "mental illness" increases the risk for bipolar disorder. Bipolar disorder is, itself, a mental illness! If the authors mean "comorbid mental illness" (such as an anxiety disorder) then this must be mentioned using appropriate terms. Otherwise, it is a serious error which reflects badly on the credibility of the researchers.

Author’s response: Thank you for your valuable observation. You are correct; it would be ‘comorbid’. We have corrected these accordingly throughout the manuscript.

4. Similarly, the authors have used simplistic, non-standard terminology ("rich" vs "poor" for income, "good" vs "poor" for quality of the marital relationship) that makes the data analyses difficult to interpret.

Author’s response: Thank you for your comment. The term rich and poor are very well known for socio-economic class measured by family income, and it has been used in different publications. Some researchers used the term good and poor for measuring status of marital relationship, and most of the researchers used the term healthy and unhealthy for marital relationship. We have revised and put healthy and unhealthy instead of good and poor respectively for marital relationship throughout the manuscript. 

5. The writing style and grammar are poor, and the article would benefit from the assistance of a writer skilled in scientific writing in English.

Author’s response: As we are not English-speaking people, we have tried our best to make our language clear, correct and unambiguous. 

6. Finally, there are serious ethical concerns about the inclusion of victims of child marriage (age 15) in a study of this sort. By including such subjects and accepting such a situation as normal, the researchers are tacitly endorsing the aberrant situation that exists in their country (https://www.independent.co.uk/news/world/asia/bangladesh-child-marriage-law-minimum-age-zero-reduce-baby-marital-unicef-un-a7619051.html) rather than advocating for the rights of vulnerable populations, as any ethical researcher would.

Author’s response: For your kind information, though the legal age at marriage for girls is 18 years still the prevalence of child marriage among girls in Bangladesh is 16.3% at 15 years, reported by Bangladesh Demographic and Health Survey. We have described with references about this issue [Page 7, Line 141-149]. 

Reviewer #2: Minor corrections:

Abstract:

1. Better to use the term “mental disorder” rather than using “mental disease”

Introduction:

Author’s response: Thank you for your constructing advice. We have replaced “mental disease” by “mental disorder” throughout the manuscript.

2. In place of the term “victim” use “patient”

Author’s response: As per your suggestion “victim” was replaced by “patient”.

Methodology

3. Authors have mentioned randomization was done at all three stages. It would be appreciated if they mention what method of randomization was used. A brief description of the process would help readers as well as future researchers.

Author’s response: Thank you for the advice. We have described the randomization process [Page 6, Line 115-124]. 

4. What does “mental position” mean in line 122.

Author’s response: It would be ‘mental state’. We have corrected it. 

5. Regarding the questionnaire, it is not clear whether the 66 items semi structured questionnaire had 20 items of BSDS or it was separately administered. Please clarify. 

Author’s response: Yes, the 20 questions of BSDS were included in the questionnaire. The total questions, 46+ 20 (BSDS) =66. We have revised to make clear [Page 6-7, Line 131-137]. 

6. For data collection you have mentioned “literate adults” were used as data collectors. I think they should have some level of education to collect data.

Author’s response: We have revised and mentioned the educational qualifications of our data collectors [Page 7, Line 137-140].

7. For female less than 18 years please mention if assent was taken.

Author’s response: Especially for less than 18 years female, we have taken their and their guardians written consent. We have mentioned in page 7 line 141-149. 

8. Grammatical error “All the information of the answered questionnaires entered in computer and coded”

Author’s response: The grammatical error has been corrected. 

Major comments:

9. In the methodology result and discussion section the authors have looks into different factors (mainly social) and have concluded the precipitating factors, vulnerability. However, the factors may have just association rather than direct precipitation. The authors have completely missed the biological and psychological factors. If not studied these could be the domains for discussion.

Author’s response: Thank you for the observation. We have looked into many factors but considered only the significant factors in our result and discussion sections. Our aim was to investigate the associated factors and we made corrections in this regard as per your advice. We did not focus on biological factors. Some psychological factors were considered.

10. Table 1: the factors like family members’ chronic illness have shown significant difference in BPD and normal groups. Similarly variable “sick treated immediately” makes no sense. This might be misleading as there is no any rational pathway for this. Hence I would recommend to take literature review in consideration before doing statistical tests and making an inference.

Author’s response: The question of ‘chronic disease’ was to know medical comorbidity, but ‘if sick treated immediately’ was to know the attention and care of the family to the woman which might affect her psychological state. 

11. How were the reference factor selected? (table 2)

Author’s response: Most of the reference factors have been selected from previous published papers. 

12. My main concern is how were the study variables selected to make inferences?

Author’s response: Most of the variables have been selected following some published papers which were related to our present study. 

We think Figure 3 is not necessary, Fig.3 has been deleted from revised manuscript. 

We would like to thank the editor and reviewers for the valuable comments. We have revised the documents to the best of our ability, but we will definitely be happy to provide further improvement if there are further clarifications required. 

With best regards

Dr. Md. Golam Hossain

Professor of Health Research Group

Department of Statistics, University of Rajshahi

Rajshahi-6205, Bangladesh

E-mail: hossain95@yahoo.com

---

## [Editor Report · Decision Letter 1]

24 Jan 2020

PONE-D-19-24745R1

Bipolar disorder among married women in Bangladesh: Survey in Rajshahi city

PLOS ONE

Dear Prof. Hossain,

Thank you for submitting your manuscript to PLOS ONE. After careful consideration, we feel that it has merit but does not fully meet PLOS ONE’s publication criteria as it currently stands. Therefore, we invite you to submit a revised version of the manuscript that addresses the points raised during the review process.

The authors have incorporated most of the comments in the revised manuscript; however, some of the writing is a bit imprecise, the manuscript be edited by a professional English language editor before publication.

We would appreciate receiving your revised manuscript by Mar 09 2020 11:59PM. To enhance the reproducibility of your results, we recommend that if applicable you deposit your laboratory protocols in protocols.io, where a protocol can be assigned its own identifier (DOI) such that it can be cited independently in the future. For instructions see: http://journals.plos.org/plosone/s/submission-guidelines#loc-laboratory-protocols

We look forward to receiving your revised manuscript.

Kind regards,

Nagendra P. Luitel, MA

Academic Editor

PLOS ONE

---

## [Author Response · Author response to Decision Letter 1]

5 Feb 2020

Response to Editor and Reviewers 

Journal Name: PLOS ONE 

Tracking No. (Manuscript ID): PONE-D-19-24745R1

Manuscript Title: “Bipolar disorder among married women in Bangladesh: Survey in Rajshahi city"

Dear Editor,

Thank you very much for providing your comments on our manuscript. We have modified and revised the manuscript accordingly, and detailed corrections point-by-point is given below:

Comment:

The authors have incorporated most of the comments in the revised manuscript; however, some of the writing is a bit imprecise, the manuscript be edited by a professional English language editor before publication.

Author’s response: Thank you very much for your useful comments on our manuscript. 

(i) According to your suggestion, we have tried to precise our manuscript. 

(ii) As you suggested, we have sent our manuscript to Professor Jyoti Sarkar, Department of Mathematical Sciences, School of Science, Indiana University Purdue University Indianapolis for improvement of language in this manuscript. He has revised English throughout the manuscript. 

We would like to thank the editor and reviewers for the valuable comments. We have revised the documents to the best of our ability, but we will definitely be happy to provide further improvement if there are further clarifications required. 

With best regards

Dr. Md. Golam Hossain

Professor of Health Research Group

Department of Statistics, University of Rajshahi

Rajshahi-6205, Bangladesh

E-mail: hossain95@yahoo.com

---

## [Editor Report · Decision Letter 2]

10 Feb 2020

Bipolar disorder among married women in Bangladesh: Survey in Rajshahi city

PONE-D-19-24745R2

Dear Dr. Hossain,

We are pleased to inform you that your manuscript has been judged scientifically suitable for publication and will be formally accepted for publication once it complies with all outstanding technical requirements.

With kind regards,

Nagendra P. Luitel, MA

Academic Editor

PLOS ONE

Additional Editor Comments (optional):

No any further comment. The authors have addressed all the comments and feedback from the reviewers and academic editor.
---

## [Editor Report · Acceptance letter]

13 Feb 2020

PONE-D-19-24745R2 

Bipolar disorder among married women in Bangladesh: Survey in Rajshahi city 

Dear Dr. Hossain:

I am pleased to inform you that your manuscript has been deemed suitable for publication in PLOS ONE. Congratulations! Your manuscript is now with our production department. 

With kind regards,

on behalf of

Dr. Nagendra P. Luitel 

Academic Editor

PLOS ONE